# Insects in Art during an Age of Environmental Turmoil

**DOI:** 10.3390/insects13050448

**Published:** 2022-05-09

**Authors:** Barrett Anthony Klein, Tierney Brosius

**Affiliations:** 1Department of Biology, University of Wisconsin—La Crosse, La Crosse, WI 54601, USA; 2Department of Biology, Augustana College, Rock Island, IL 61201, USA; tierneybrosius@augustana.edu

**Keywords:** art, climate change, colony collapse disorder, cultural entomology, environmental art, ethnoentomology, habitat destruction, insect art, invasive species, pollution

## Abstract

**Simple Summary:**

The diversity of life on Earth is declining due to human decisions and human actions. Scientists have clearly identified categories of human-induced environmental distress, and public awareness is growing, yet science and related media reports are not affecting enough policy change to forestall our impact. Additional approaches need to be taken, and one potent vehicle for eliciting responses is art. Some visual artists have chosen to include insects in their work. Insects are diverse, abundant, ecologically and culturally important to us, and are suffering declines by our hand. These qualities, coupled with insects’ uncanny ability to evoke emotional extremes, marks them as uniquely powerful subjects for artists to convey messages about our relationship with the planet. We surveyed relevant work by 73 artists and found a bias favoring insect art addressing habitat destruction or climate change, and an underrepresentation of art related to several other important categories of environmental destruction. Art favored Hymenoptera over all other insect orders, including orders containing more described species. Noting these misalignments, we see opportunities for artists to more extensively explore insect diversity and the harm we are causing, and for art to increasingly play a complementary role in affecting change in our destructive behavior.

**Abstract:**

Humans are reshaping the planet in impressive, and impressively self-destructive, ways. Evidence and awareness of our environmental impact has failed to elicit meaningful change in reversing our behavior. A multifaceted approach to communicating human-induced environmental destruction is critical, and art can affect our behavior by its power to evoke emotions. Artists often use insects in their works because of our intimate and varied relationship with this diverse, abundant lineage of animals. We surveyed work by 73 artists featuring insects or insect bodily products to gauge how extensively artists are addressing anthropogenic environmental distress, and what insects they are choosing as subjects in the process. Categories often cited as contributing to species extinction are (1) habitat destruction, (2) invasive species, (3) pollution, (4) human population, and (5) overharvesting. After adding insect-specific categories of (6) decline of insect pollinators and (7) the intentional modification or extermination of insects, we categorized our surveyed works, confirming categorizations with 53 of the living artists. Forty-seven percent of the artists addressed habitat destruction or climate change, but some other categories were severely underrepresented, with almost no work explicitly addressing overpopulation or overharvesting. Artists favored Hymenoptera (62%) over potentially more species-rich orders. Recognizing these biases could alert scientists, artists, and others to more effectively communicate messages of universal importance.

## 1. Introduction

“We stand guard over works of art, but species representing the work of aeons are stolen from under our noses”.—Aldo Leopold

The human aptitude for modifying environments has become a hallmark of our species. As a result of this talent, we have doubled our average life expectancy in the last 200 years [1]. Over the last 50 years, the global economy has grown nearly fourfold [2] and extreme poverty has declined by over 50% [3]. The successful altering of environments and corresponding advances in medical technology has resulted in an explosion of the global human population to almost 8 billion [4]. As remarkable as these developments are, they have come at a frightening ecological cost. Drastic changes in human population and resulting economic growth have increased demand for energy and materials at an alarming rate. Land is cleared for food and extraction of resources [5], accidental and intentional introductions of organisms to new locations have permanently destabilized entire ecosystems [6,7,8], and the sharp increase of atmospheric CO_2_ levels has resulted in global temperature rises that have been connected to unprecedented heat waves, droughts, and other extreme weather events [9,10,11].

Several recent studies raise red flags regarding what is being called the sixth mass extinction event [12,13,14,15,16,17]. Ocean ecosystems are under increasing pressure from climate change [18,19], and declines in nearly every major vertebrate group have been recorded [20,21]. Terrestrial invertebrates also appear to be declining [13,22,23,24,25]. Insect populations are not inexhaustible, and a reckoning based on this realization has been captured with headlines such as “The Insect Apocalypse Is Here” [26] and “An Insect Apocalypse Will Be Our Apocalypse” [27]. As we are hit with waves of studies documenting biodiversity loss throughout the world, it is clear that humans are the primary cause of climate change and rapid environmental destruction [28]. There is nearly universal acceptance among experts that the planet’s atmosphere has a growing abundance of CO_2_ and other greenhouse gases as a result of burning fossil fuels [29,30]. The resulting loss of biodiversity caused by human behavior will alter the functioning of the ecosystems that we rely on for future prosperity [31]. E. O. Wilson adopted the acronym “HIPPO” as a way to summarize threats causing biodiversity loss: H = Habitat loss, I = Invasive species, P = Pollution, P = human Population, and O = Overharvesting. The threats are not independent of each other, and human population affects all other threats, but if this second P in HIPPO is removed, Wilson suggests that HIPPO represents the threats in order of decreasing magnitude, at least in the best known taxa (including vertebrates and flowering plants) [32]. While it is clear that there is a problem and we are the primary cause of this problem, there is no consensus on how to reverse course. As more time passes and no substantial actions are taken, damage to our planet continues to increase [33]. Human-induced environmental destruction is the most important issue of our time, and having a dangerously disjointed leadership makes it difficult to imagine a unified effort to curtail it [34].

The lack of motivation to stop environmental destruction might hint at a general ignorance of how our actions are affecting the planet. This does not seem to be the case, however. Surveys suggest the public is aware of climate change [35,36]. Environmental destruction is featured prominently through many news outlets and is taught at different levels of education, including primary school. It seems that while news coverage does lead to awareness of problems with the environment, it does not directly translate to public engagement or policy acceptance [37]. To enact change, it clearly is insufficient to rely on empirical evidence and scientific reports alone, or on media sources to report on these findings.

Behavioral decision research suggests that worry drives perceptions of risk and lack of emotional response leads to inaction [38]. When people fail to perceive risk, they do not take action. The very nature of environmental destruction, especially climate change, is often abstract and time-delayed, which leads to a decreased perception of risk [38,39,40,41]. Even if we are successful at increasing concern about climate change, we may find that tactic to be ineffective. The current messaging around environmental destruction, including climate change, has often been fear. While fear can grab attention and motivate action [39], can it motivate long-term change? Current research suggests that this type of messaging may be ineffective or even counterproductive [42,43,44]. The wide range of emotions associated with environmental destruction and our response to those emotions is complicated. Emotions such as fear, anger, guilt, and even pride have all been connected to environmental motivation [44,45,46,47]; however, we need to understand how best to nurture these emotional responses in a constructive way that has a meaningful impact.

A growing number of individuals are turning to art to help raise public awareness of topics related to environmental destruction [40,41,47,48]. Art’s ability to evoke emotions while encouraging dialogue may serve as a powerful tool in communicating the importance of the environment. Art has been found to facilitate discussion among stakeholders and to increase group discussions [40,47,49,50,51,52]. Using a survey of 874 spectators of art that accompanied the 21st UN climate summit in Paris, Sommer and Klöckner found that climate change-related art may serve as an effective way of triggering emotional responses [51].

As entomologists, we were particularly interested in the power of insects to serve as influencers on the topic of environmental awareness. Insects are commonplace and familiar to everyone, and insect imagery elicits a wide range of emotional responses. For some cultures, insects have become symbols and are subjects to be celebrated, yet for others insects can be the objects of intense fear, potentially with deep roots in our evolutionary psychology [53]. Can artists use this combination of familiarity and uneasiness we have towards insects to make unique contributions to the growing and influential movement of environmental art? Are insects particularly suited for communicating some aspects of environmental destruction? We surveyed examples of insect art—art featuring insects [54,55,56,57,58] or incorporating insect bodily products [59,60,61]—that addresses human-induced environmental destruction to see how prevalent this body of art is, and if biases exist with respect to artists’ attention to categories of destruction or taxa of insects.

## 2. Materials and Methods

We conducted a survey of art to find any work using insects that addresses the topic of anthropogenic environmental destruction. To identify works that specifically involve humans as drivers of environmental distress or biodiversity loss, we surveyed:Collections of insect-themed art, published in books and exhibit catalogs (see references for partial list);Insect-themed gallery exhibits, accessible online;Articles written about artists using insects in their work, found online or from B.A.K.’s personal collection of tangible and digital files;Books about environmental art (see references for examples);Social media, by making calls for thematically relevant examples through Facebook and Twitter;Artists, by asking for examples of others’ art we had not already listed.

Humans are impacting the planet in different ways. We created categories that addressed widely cited drivers of human-induced environmental distress, with some focus paid to more insect-specific issues. Borrowing from the common mnemonic HIPPO, encapsulating causes of species extinction [32], and categories we created for a book chapter in a series about cultural entomology [62], we created the following categories for this treatment:Habitat destruction/change, including climate change;Invasive species;Pollution, including use of pesticides;Human population;Overharvesting by hunting;Decline of pollinators, including colony collapse disorder (despite human involvement not being clear with regard to colony collapse disorder);Intentional modification (e.g., bioengineering) or extermination of insects, with concern for insects or the environment in mind;Concern for environment/insects (when human involvement is not made explicitly clear).

The first five categories are slightly modified from HIPPO. We created the final category in case we found art that appeared to us to be perfectly relevant, but we could not confirm in which category, if any, the works belong. We included artists in our survey if descriptions of at least one of their insect works was explicitly relevant to the theme, as expressed by the creators of the art, or by authors, journalists, or critics writing about the art. If the artist produced multiple relevant works, we selected representative pieces that maximized the number of different categories of human-induced environmental distress or the number of insect orders featured. Work was relevant if an artist was motivated to produce an insect work to address one of the categories listed above or wished to exhibit the work to convey a message relevant to one of these categories. Artists often describe their motivations for creating specific pieces, or an entire body of their work on their personal websites, in interviews, or in exhibition catalogs. Secondhand accounts, written by others, frequently express artists’ motivations, or attempt to independently interpret artists’ works. When possible, we contacted each artist directly to confirm that our categorization of their work was appropriate. When contact was made, we replaced secondary accounts (not made by the artists) with the artists’ own interpretations of their work. In several cases, we asked artists if their insect art was relevant to the list of categories, if we suspected that it might be but had found no evidence elsewhere to confirm this.

## 3. Results

Environmental art is typically modern or contemporary, so given the nature of our survey, none of our results feature insect art predating the recent environmental movement. Our survey includes 73 artists, or teams of artists, who have produced at least one art piece that features insects or insect bodily products relevant to this article’s theme of human-induced environmental distress. Each of our approaches to finding artists was helpful, though use of social media generated the fewest examples of relevant art. We attempted to contact all living artists (at least two were dead at the time of writing), and 53 (73%) of the artists were able to fact-check our information about their work. Five of these artists produced works relevant to these categories, but we had no evidence to confirm that their intentions were aligned with this article’s theme until we contacted them (“Ref” cell empty and “pc” cell filled in Table A1).

### 3.1. Categories of Destruction

Artists created works unequally across the categories of environmental concern. Most artists produced insect art relevant to habitat destruction or climate change (34 of the 73 artists; 47%; Figure 1), followed by pollution (23 artists; 32%; Figure 2), decline of pollinators or colony collapse disorder (19 artists; 26%; Figure 3), invasive species (13 artists; 18%; Figure 4), and the intentional modification or extermination of insects (10 and 3 artists, respectively; 18%; Figure 5). We found only one example of insect art where the artist addresses human overpopulation, and one work where the artist addresses overharvesting by hunting, and in both of these cases other categories were cited as complementary concerns. Fourteen (19%) of the artists produced works that expressed concern for insects or the environment without citing specific human involvement, or listing other categories of concern (Figure 6). Though concern for the environment or insects was considered implicit for works assigned to the other categories, five of the artists wished to include an additional, specific category of concern (not included in calculations, but listed in Table A1).

Each artist is maximally represented five times in the above calculations if their work was relevant to five of the categories (*n* = 2), though most artists are represented only once (45 of the artists; 62%), or twice (18 of the artists; 25%), limiting a single artist’s bias on the results (Figure 7).

If we exclude artists who addressed more than three categories, the relative proportions of artworks conveying messages related to the categories does not markedly change (Table 1).

### 3.2. Insect Orders

Of the 73 artists, two artists’ works were performances that involved any and all insects that opportunistically appeared on location, and three other artists have featured an unknown number of insect orders across a series of works. The remaining 68 artists collectively showed biases when selecting insects to include in their works related to anthropogenic environmental change. Hymenoptera were the overwhelming favorites (*n* = 42 of the 68 artists; 62%), with the closest contenders being Lepidoptera and Coleoptera (24 of the artists featured each order; 35%), followed by Hemiptera (17 artists; 25%) and Diptera (12 artists; 18%).

The same hierarchy of orders exists when analyzing the 148 records of orders in our survey, starting with Hymenoptera (*n* = 42; 28%) and Lepidoptera and Coleoptera (24 artists featured each order; 16%). Most artists featured only one (*n* = 42; 62%) or two (*n* = 8; 12%) insect orders in our survey, but two artists’ works feature nine insect orders. A single artist, or team of artists, using a great number of insect orders in their work could have an outsize influence on the results, so after excluding art featuring more than three orders of insects (“Mix” in Table 1), we found the ordinal bias did not dramatically change, with a bias still favoring Hymenoptera (Table 2).

Hymenoptera appeared in more artworks than any other order—overall, and within several of our categories related to anthropogenic environmental destruction. Hymenoptera appeared in the most works addressing invasive species (*n* = 8; next closest: *n*_Coleoptera_ = 7), pollution (*n* = 13; next closest: *n*_Lepidoptera_ = 5), decline of pollinators or colony collapse disorder (*n* = 20; next closest: *n*_Lepidoptera_ = 5), and intentional modification of insects (*n* = 5; next closest: *n*_Diptera_ = 4). Habitat destruction or climate change was strongly represented by two other orders (*n*_Hymenoptera_ = 22 vs. *n*_Lepidoptera_ = 20 and *n*_Coleoptera_ = 23; Figure 8). The categories of human overpopulation and overharvesting by hunting have almost no examples, so a consideration of ordinal bias is beyond speculation.

## 4. Discussion

Artists featuring insects in work addressing anthropogenic environmental distress did so primarily as it relates to habitat destruction or climate change, followed by pollution, decline of pollinators (or concern about colony collapse disorder), invasive species, and the intentional modification or extermination of insects. Only a single work was related to the issue of human overpopulation or to overharvesting by hunting (Figure 9), at least explicitly by each artist, and in each of these cases other categories were cited as complementary concerns (Table A1). If our categories of anthropogenic environmental distress are a reflection of threats to biodiversity, and E.O. Wilson was accurate when listing the most destructive forces causing biodiversity loss in hierarchical order using HIPPO (noting caveats mentioned in the Introduction), then our surveyed art shows an underrepresentation of works relevant to the category of invasive species, and a nearly complete dismissal of or neglect to explicitly address the threats of human overpopulation or overharvesting by hunting.

What leads to the misalignment of the magnitude of these environmental threats? There is a large body of research devoted to risk perception among the general public and how perceived risks and actual risks often do not align [38,63]. Most artists are not scientists, and they consume the same news media and are influenced by the same factors generating risk perception as other members of the general public. Insect art addressing invasive species may be less prevalent than expected because invasive species’ impact can be more difficult to perceive. Much like climate change, the damage wrought by invasive species can take years to noticeably manifest [38]. When artists did address invasive species in their insect art, their motivations varied greatly, and included whimsy based on insect name, insects themselves being threatened by non-insect invasives, insects innocently taking advantage of ecosystem imbalances caused by humans, and visions of invasive insects as threats to native species (Figure 4 and Figure 8). As for addressing human overpopulation, examples of insect art may appear severely limited for a variety of reasons. Overpopulation is a difficult topic to discuss, and conversation related to curbing population growth can be fraught with controversy. Overpopulation exacerbates all of the other categories of concern, so perhaps artists are addressing more productive conversations connected to unsustainable population growth. Consumption of resources, reducing plastic waste, and female empowerment through education [64] are all part of the larger body of environmental activism. Even the single piece that fit within this category in our survey addressed the complexity by also relating to three other categories.

Our survey included categories in addition to HIPPO and more specific to insects. We found that artists are actively producing works relevant to the decline of pollinators (including concern about colony collapse disorder), and to the intentional modification or extermination of insects. Art motivated by concern for insect pollinators can be subtle, or explicit and haunting; in *Traces* (2015), for example, Beate Kratt portrays the ominous “essence of the last bee’s dance captured in a jar” [65]. There is broad concern for honey bees, but many artists surveyed were mindful of a wealth of other insect pollinators. Examples include creative works of activism sculpted to aid pollinators in need (Figure 3). Works addressing the intentional modification of insects often allude to genetic modification (Figure 5), but also to industrialized farming or other related topics. Though three artists addressed the intentional extermination of insects, only one artist in our survey created works that do so exclusively, and the works are appropriately stark and dire (Catherine Chalmers’ *Executions*, 2003).

Just as artists exhibited a bias with respect to category of destruction, artists also chose their six-legged subjects with taxonomic bias. The most diverse insect orders are highly represented in our survey, but the number of artworks surveyed did not perfectly match the species diversity described within these orders (Figure 10). Hymenoptera were the overwhelming favorites, with Coleoptera being grossly underrepresented and Hemiptera being somewhat inflated in their representation. While there is some evidence that Hymenoptera could surpass Coleoptera as the most speciose order of animals on the planet [66], currently the number of described beetle species far surpasses that of Hymenoptera. The Hymenoptera bias is likely explained by a combination of factors centering around our ancient and positive associations with a single species—the western honey bee (*Apis mellifera* Linnaeus, 1758), which appears far more often than any other species in our survey. Historical associations with honey bees and beekeeping [67,68], love of honey, use of beeswax in encaustics [69], reliance on honey bees for pollination, and concerns about colony collapse disorder are just a few of the reasons why artists may turn to Hymenoptera above other insect orders. Hymenoptera not only supports a great diversity of insects, but a tremendous abundance of insects. Abundance of insects within orders, or biomass of insects per order may also affect artists’ choices, though a lack of data on insect ordinal abundances prevents a careful assessment of this relationship.

### 4.1. Future of Insect Art Addressing Anthropogenic Environmental Distress

Humans’ and insects’ lives have long been deeply intertwined. Insect diversity and abundance, the ecological services insects perform, and the cultural connections [71,72,73,74,75] we share spell a future of continued reliance and inspiration. Perhaps the use of insects presents a novel way of helping humanity grasp the magnitude of climate change. Thinking creatively about our environmental problems can emerge from the same processes associated with thinking about art [76]. Creativity and innovation have resulted in human-centric advances, and it is possible that this creativity and innovation will affect positive environmental change. Environmental art has a history of shaping debates, and our present environmental crisis only adds urgency to environmental artists’ practice [77]. Our survey exposes unique and creative ways artists incorporate insects when conveying messages about our treatment of the planet, and we expect the frequency, variety, and innovation of insect art relevant to this theme to expand as our crisis deepens.

Just as we cannot rely on an accumulation of scientific data to turn our crisis around, art will not, by itself, solve our environmental problems. The artist Mark Dion astutely pointed out that “to build a culture of nature that features regeneration over destruction, sustainability over depletion and nurturing over domination, it requires input from a diverse collation of thinkers, makers, and doers. Art is one of many areas which can be important to this constellation.” We are faced with a difficult task that will take a globally coordinated effort to maintain life’s diversity. Coordinated effort requires openly sharing information, with platforms optimized for dissemination to a diverse audience. Many scientists are speaking about their work directly using social media platforms, and this can help to make science more accessible to people outside of the scientific community. People need to feel empowered to take action, and part of being empowered is being informed. An educated populace can make informed choices and vote in ways that reflect their understanding of crucially important issues. As scientists who are acutely familiar with the challenges facing us in the coming decades and as entomologists who are enamored and fascinated with the beauty and biology of insects, we find hope and optimism when insects are used creatively as vehicles to communicate the most globally important issues of our time.

### 4.2. Qualifiers of Survey

It is possible that our results are tainted by sampling error, and that a multitude of insect artworks exists relevant to human overpopulation, or overharvesting, or the pattern of insect orders featured in relevant works more closely resembles the relative species diversity within those orders. Our survey suffers from multiple biases. Our search for relevant artworks was constrained by what art and interpretations of art were publicly available, or were privately communicated. Not all artists are explicit about their motivations or intentions. We were fortunate in that 53 of the 73 artists confirmed or modified our categorizations of their work, but there is an unknown bounty of works we were unable to identify as relevant to our survey theme. This includes works by artists who are dead and did not convey their motivations, or by living artists who keep their intentions private. The environmental art movement is relatively recent, but individual artists’ concerns for biodiversity or concerns about our mistreatment of the planet may predate this movement, and older, relevant insect art may exist. Our search is geographically and culturally biased, in large part because of personal language barriers, but also because of limited access to works not widely exhibited, circulated, or posted online. Our request by social media for relevant works was constrained by our personal or professional social network, which does not reach extensively into communities in Africa, Asia, or Central and South America. There are without doubt artists throughout the world concerned about humans’ impact on the planet who express their concerns by featuring insects. We will continue to uncover their works.

## 5. Conclusions

Artists motivated to convey messages related to anthropogenic environmental distress have unique opportunities when featuring insects as their subjects. We found that artists have categorically explored different ways in which humans harm biodiversity, but with biases that do not perfectly reflect the relative severity of each of these categories. Insect art addressing habitat destruction, including climate change, are appropriately most common, but we might expect more work to address invasive species, and more than the single examples we found addressing human population growth and overharvesting. Further, artists exhibited a Hymenoptera bias in works relevant to this study, with more speciose orders of insects (Coleoptera, Lepidoptera, Diptera) less frequently represented. Artistic biases exist here for many reasons worth acknowledging. We suggest that art can serve a unique role, complementing public education and scientific and media reports, to elicit change in our behavior and our environmental policies.

## Figures and Tables

**Figure 1 insects-13-00448-f001:**
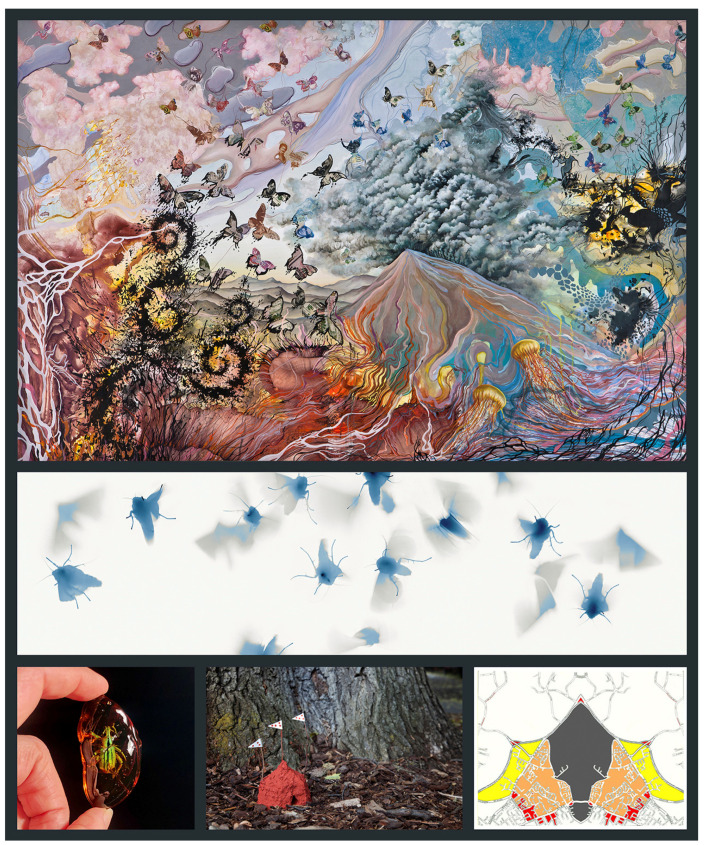
Insect art addressing habitat destruction or climate change. Butterflies in *Melt* (**top**; acrylic, ink and collage; 2012) are converted into monetized creatures and speak to the effects of economic power and devastation. Erika Harrsch: “I created [*Melt*] after I saw from the plane flying over the north pole all the detached fragments of melting ice.” *Moth Liturgy 1* (**middle**; pigment inkjet print from digitally modified scans of gelatin silver film photograms, 2016) is from Harry Nankin’s series featuring live Bogong moths, *Agrotis infusa* (Boisduval, 1832), from an ecosystem in the Australian Alps “doomed by anthropogenic climate change.” A weevil (**bottom left**; in resin, 2020) from Jenny Kendler’s *Amber Archive*. Perdita Phillips created *Termite Embassy* (**bottom center**; papier-mâché and cardboard, 2015) in response to the Paris Climate Accords. A butterfly emerges from a modified and mirrored map of Great Hollands, an area that will metamorphose over time “to accommodate our growing population” in *Surrey Butterflies* (**bottom right**; Angela Thames; 2007–2008). All images are courtesy and copyright of the artists.

**Figure 2 insects-13-00448-f002:**
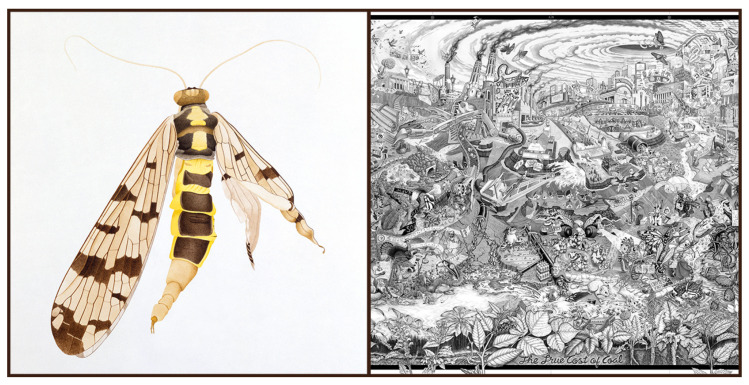
Insect art addressing pollution. Cornelia Hesse-Honegger has artistically and scientifically investigated and documented deformed insects within the vicinity of nuclear power plants for decades. *Scorpionfly, Panorpidae* (**left**; aquarell, 1988) records wing and abdomen deformities of a scorpionfly from Reuental, near nuclear power plant Leibstadt (image courtesy and copyright of the artist). In contrast to highlighting a single specimen on white, The Beehive Design Collective depicts multiple human impacts, including pollution’s effects on peppered moths (near smoke stacks), a caddisfly, and many others in *The True Cost of Coal* (**right:** pen and ink, 2010; Creative Commons). Inspired by the back cover of Mad Magazine, this is only the central portion of a larger fold-out poster, featuring mountaintop removal and climate chaos.

**Figure 3 insects-13-00448-f003:**
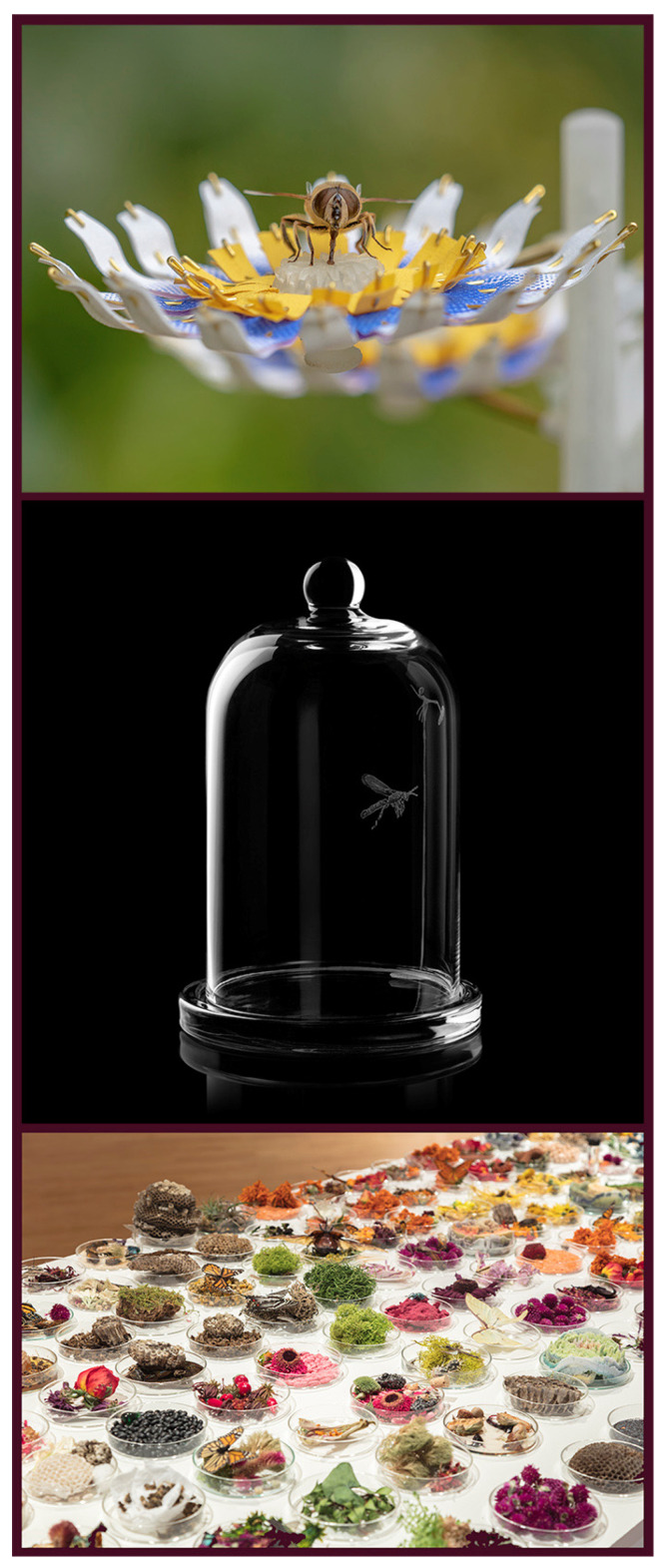
Insect art addressing the decline of pollinators, or concern about colony collapse disorder (CCD). Matilde Boelhouwer has worked with scientists and engineers to develop a series of artificial flowers to serve as “an emergency food source for the ‘big 5 of pollination.’” Here (**top**), a syrphid hover fly feeds from one of these flowers in *Insectology: Food for Buzz* (2018). In *Threatened, Rare**—Extant* (**center**; Susan Hauri-Downing; 2018), a glass dome contains an engraved image of *Zaspilothynnus gilesi* Turner, 1910, a thynnine wasp that pollinates an endangered orchid, threatened by a battery of destructive acts executed by humans. Specifically addressing CCD (and bio-engineering), Suzanne Anker’s *Twilight* (**bottom**; 2016) includes pollinators among other natural items in Petri dishes. All images are courtesy and copyright of the artists.

**Figure 4 insects-13-00448-f004:**
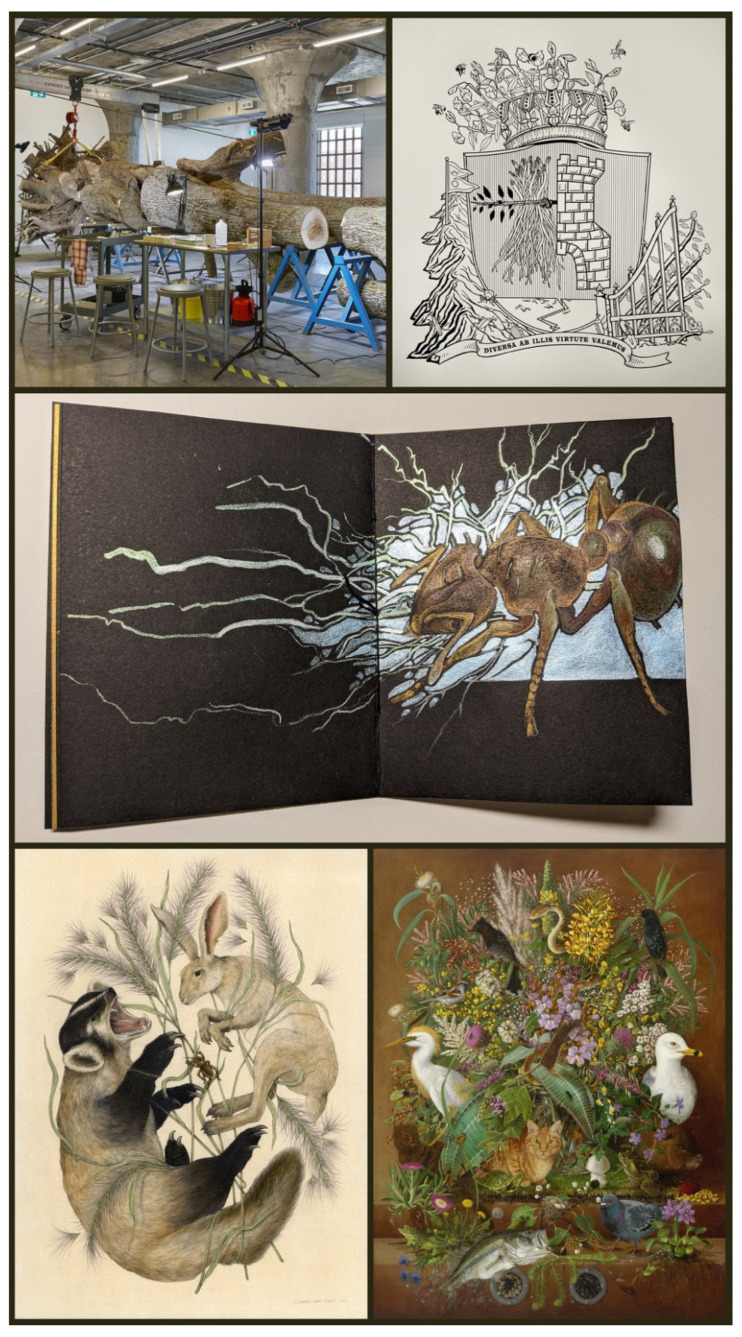
Insect art addressing invasive species. The emerald ash borer (*Agrilus planipennis* Fairmaire, 1888) is the invasive species in *Life of a Dead Tree* (**top left**; 2019), in which Mark Dion worked with entomologists to collect insects from the felled, 140-year-old tree. Marina Zurkow’s *Heraldic Crests for Invasive Species* series features the invasive Himalayan Balsam (**top right**; letterpress print, 2011), with two beetles as its enemies: *Acropteroxys gracilis* (Newman, 1838) and *Mecinus janthinus* Thomson, 1865. Insects “are not the invaders; rather, they’re taking advantage of anthropogenically caused ecosystem imbalances.” A lighter (but still electric) approach is taken by Karen Anne Klein in her *Invaders* handmade book series (**middle**; *Electric Ants*; color pencil and inks, 2022). A lone Jerusalem cricket (*Stenopelmatus monahansensis* Stidham and Stidham, 2001) floats between badger and jackrabbit in Elizabeth Jean Younce’s *A Moment so Rare* from *The Withering* series (**bottom left**; graphite and watercolor, 2021). The insect is rare in an ecosystem overrun by the invasive Buffelgrass. Twelve invasive species of insects populate *Ascendant* (**bottom right**; Isabella Kirkland; oil and alkyd, 2000; for key to species see https://www.isabellakirkland.com/; accessed on 5 May 2022). All images are courtesy and copyright of the artists; Zurkow’s image is also courtesy of bitforms gallery.

**Figure 5 insects-13-00448-f005:**
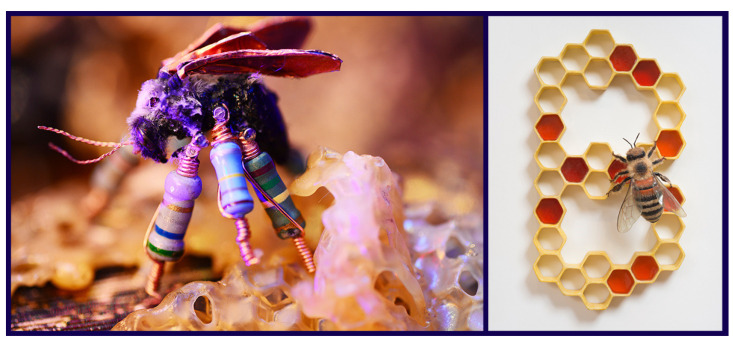
Insect art addressing the intentional modification of insects. The dramatically modified cyborg of a bumblebee specimen (**left**; *Cyberhive*; 2019) contributes to Ruth Marsh’s “wry, dystopic vision of a future wherein all bees have perished due to human causes.” Marsh displays bee specimens “repaired” with discarded electronics, and animates them for short films. Victoria Fuller imagines the product of our genetically modifying a honey bee to spell in *Spelling Bee* (**right**; craft fur, epoxy clay, acrylic, resin, Mylar, Chloroplas; 2014). Both images are courtesy and copyright of the artists.

**Figure 6 insects-13-00448-f006:**
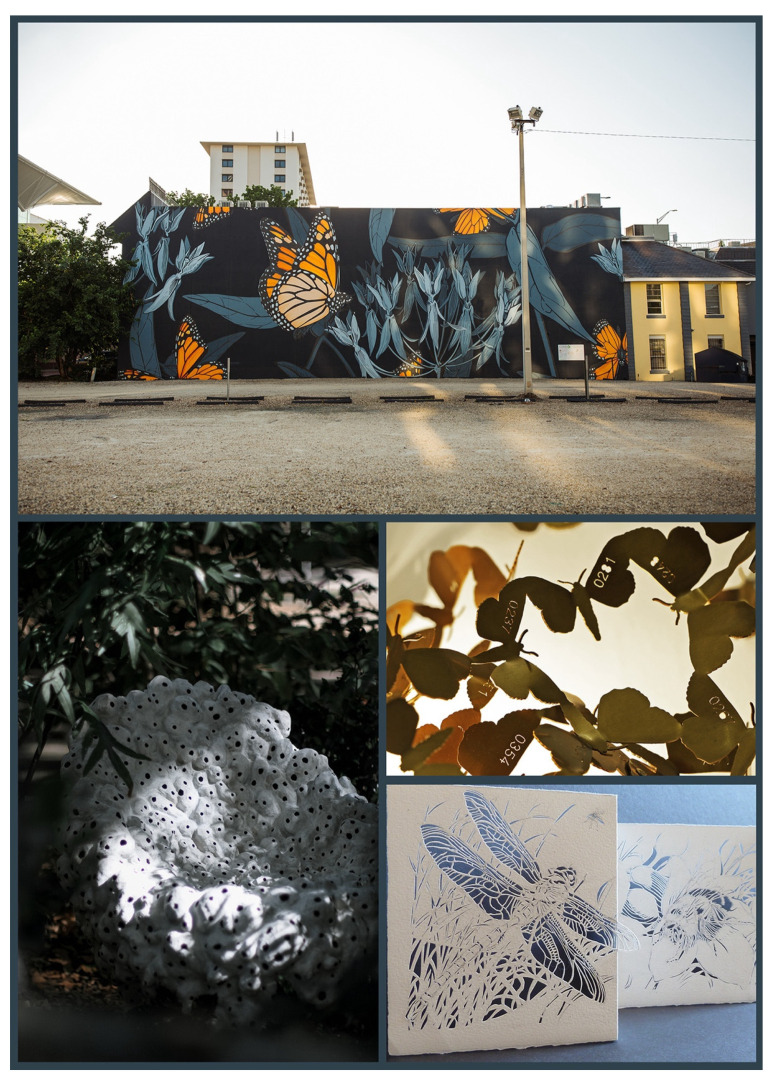
Art expressing concern for the insects or the environment, when human involvement was not made explicitly clear. One of Jane Kim’s monumental monarch murals in her *Migrating Mural* series, *Midnight Dream* (**top**; 2018) spans 3500 square feet in downtown Orlando, Florida, directly across from City Hall. Marlène Huissoud, also concerned about insect pollinators, has created a series of functional sculptures, *Please stand by* (**left**; natural clay and binder, 2021), within which urban pollinators can find shelter. Katharina Mischer and Thomas Traxler (mischer’traxler studio) have created a series of installations of metal moths appearing to fly near lit bulbs, each bulb attracting a different species of moth. Each metal moth is numbered and represents one moth remaining in Austria (*limitedMoths*; ongoing since 2008). These moths (**middle right**) represent *Catocala conversa* (Esper, 1783) in patinated brass. Vera Ming Wong created *Airborne* (**bottom right**; cut paper, 2012) with concern “about the myriad assaults by humans upon insects and other invertebrates.” All images are courtesy and copyright of the artists.

**Figure 7 insects-13-00448-f007:**
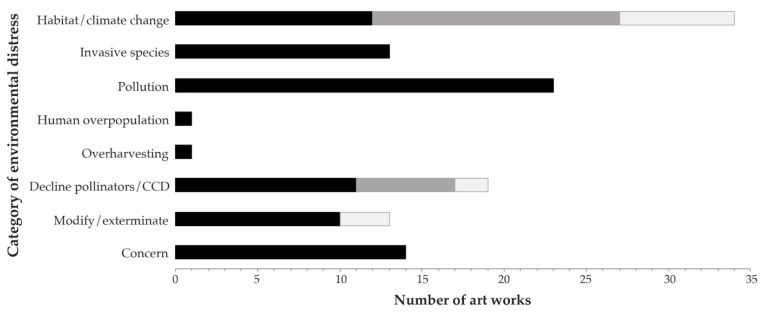
Number of artworks related to categories of human-induced environmental distress. Each artwork potentially relates to more than one category of environmental distress, but each category is calculated here no more than one time per artist (*n* = 118 category associations across 73 artists). CCD = colony collapse disorder. Modify/exterminate refers to the intentional modification or extermination of insects. Concern indicates general concern for the environment or for insects, when a more specific categorical assignment could not be made. Black bars represent the first component for each category (e.g., habitat destruction), and light grey bars represent the second component (e.g., climate change). Dark grey bars signify a combination of the two (e.g., habitat destruction and climate change were both cited as relevant to the artwork). From top to bottom, the first five categories were largely adopted from what E.O. Wilson considered the top causes of species extinction, encapsulated in the acronym HIPPO [32].

**Figure 8 insects-13-00448-f008:**
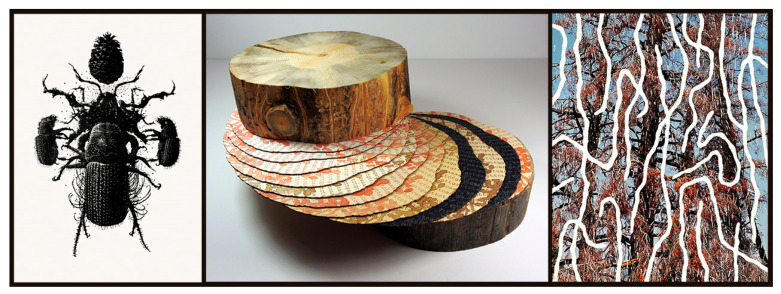
Bark beetles as vehicles for addressing environmental destruction. Certain species or lineages of insects appeared repeatedly in our sample. Although not as common as the western honey bee or monarch butterfly, bark beetles feature in works by, from left to right, Tim Musso (*Rite of the* Dendroctonus jefferyi; wood engraving, 2012), Suze Woolf (*Survivorship* (*Volume XXVII*); log with mountain pine beetle galleries, conifer mRNA texts and inked galleries on pages, 2019), and Catherine Chalmers (*Douglas Fir—Douglas Fir Beetle*; wood block print on photograph, 2022). To all three artists, damage by bark beetles indicates mismanaged forests (habitat destruction in the form of clear-cutting and fire suppression) and climate change, resulting in historically unprecedented advances by the beetles. Musso includes invasive species among the categories his works convey, due to these advances. Images courtesy and copyright of the artists.

**Figure 9 insects-13-00448-f009:**
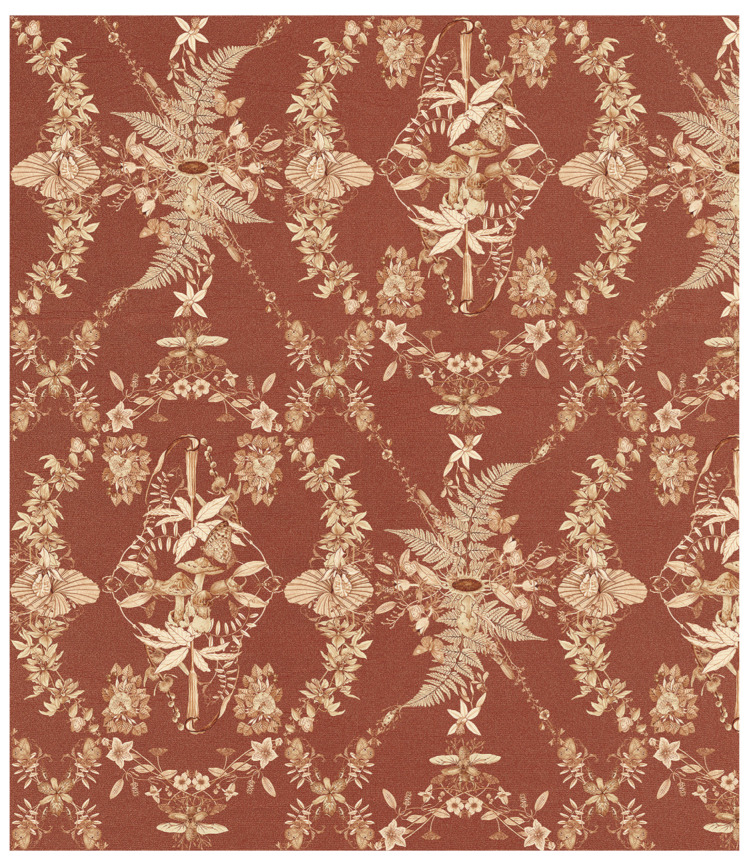
Insect art addressing overharvesting by hunting, as well as habitat destruction and climate change. Asuka Hishiki’s *Red list wallpaper KYOTO 2015* (detail of inkjet print, 2021) is a pattern comprised entirely of insects, plants, and fungi, all on the endangered red list in Kyoto prefecture, as of 2015. Included are 12 species of beetles, 3 species of butterflies, a dragonfly, a fly, and a bee, some overhunted because of their beauty. To Hishiki “This piece is a reminder to myself that we can be blind to the devastating problem. Like ‘wallpaper’, it is there in front of us, but we tend to ignore it as vague background decoration.” Image courtesy and copyright of the artist.

**Figure 10 insects-13-00448-f010:**
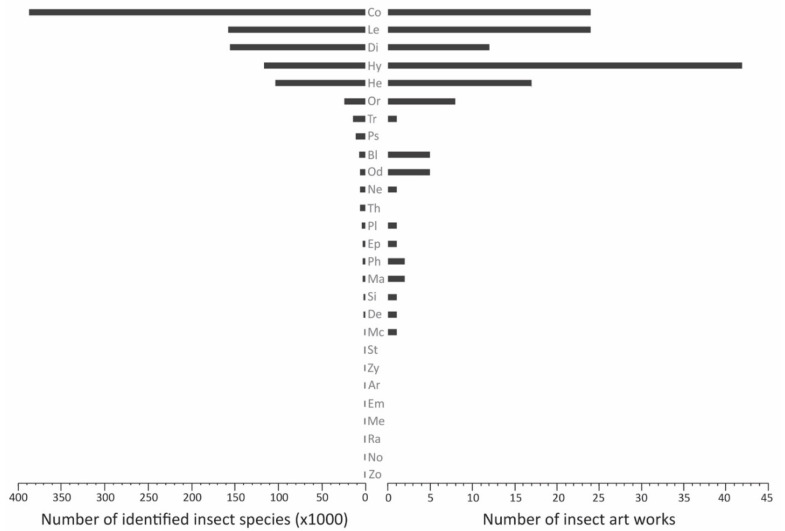
Numbers of surveyed artworks that address human-induced environmental disturbance, and total number of insect species known, organized by insect order. Each artwork potentially features more than one insect order, but each order is calculated here no more than one time per artist. Data here represent 68 of the surveyed artists (whose art has a discrete set of known, identifiable insect orders; *n* = 148 order associations). Ordinal abbreviations: Ar = Archaeognatha, Bl = Blattodea (including Isoptera), Co = Coleoptera, De = Dermaptera, Di = Diptera, Em = Embioptera, Ep = Ephemeroptera, He = Hemiptera, Hy = Hymenoptera, Le = Lepidoptera, Ma = Mantodea, Mc = Mecoptera, Me = Megaloptera, Ne = Neuroptera, No = Notoptera, Od = Odonata, Or = Orthoptera, Ph = Phasmida, Pl = Plecoptera, Ps = Psocodea, Ra = Raphidioptera, Si = Siphonaptera, St = Strepsiptera, Th = Thysanoptera, Tr = Trichoptera, Zo = Zoraptera, Zy = Zygentoma. Species numbers and order names adopted from Stork [70], though we collapsed Phthiraptera and Psocoptera into Psocodea, and Mantophasmatodea and Grylloblattodea into Notoptera.

**Table 1 insects-13-00448-t001:** Number and percentage of artworks related to categories of human-induced environmental distress. Each artwork potentially relates to more than one category of environmental distress, but each category is calculated here no more than one time per artist. We surveyed 73 artists (*n* = 118 category associations), and then surveyed 68 of the artists whose work addresses ≤ 3 of these categories (*n* = 97 category associations), to reduce bias from any artist whose work addresses many categories. From top to bottom, the first five categories relate to HIPPO [32]. CCD = colony collapse disorder. Concern indicates general concern for the environment or for insects, when a more specific categorical assignment could not be made.

	Art		Art with ≤ 3 Categories	
Category of Environmental Distress	#	%	#	%
Habitat/climate change	34	47	29	43
Invasive species	13	18	11	16
Pollution	23	32	18	26
Human overpopulation	1	1	0	0
Overharvesting by hunting	1	1	1	1
Decline of pollinators/CCD	19	26	15	22
Intentional modification/extermination	13	18	9	13
Concern	14	19	14	21

**Table 2 insects-13-00448-t002:** Number and percentage of artworks addressing human-induced environmental disturbance, organized by insect order featured in the artists’ works. Each artwork potentially features more than one insect order, but each order is calculated here no more than one time per artist. Data here represent 68 of the surveyed artists (whose art has a discrete set of known, identifiable insect orders; *n* = 148 order associations), and then 57 of the artists whose work addressed ≤ 3 insect orders (*n* = 80 order associations), to reduce bias from any artist whose work features many insect orders. Only the eight insect orders most frequently featured are included here.

	Art		Art with ≤ 3 Orders	
Insect Order	#	%	#	%
Hymenoptera	42	62	32	56
Lepidoptera	24	35	16	28
Coleoptera	24	35	13	23
Hemiptera	17	25	6	11
Diptera	12	18	6	11
Orthoptera	8	12	2	4
Blattodea	5	7	2	4
Odonata	5	7	1	2

## Data Availability

Data are archived and are publicly available here: https://osf.io/8fdyu/ (accessed on 5 May 2022).

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
