# Peer review of "Insects in Art during an Age of Environmental Turmoil"

_insects, 2022, doi:10.3390/insects13050448_

Round 1

Reviewer 1 Report

In the past, insects appearing in art of Medieval Europe and ancient Egypt have been often studied (for example, Nazari (2014) and Nazari & Evans (2015)). In contrast, analysis subject of this paper is mainly modern art. Generally this paper has high originality.

Authors can cite important references and offer appropriate discussion, tables, and figures.

As authors mentioned in Line 501-504, many arts which are listed are produced in Western Culture. Especially, China is a historic artistic big power, but Asian arts are not discussed much. This bias is a weak point of this paper. It is desirable to do comparative analysis between Western and other regions if possible. However, language barriers cannot be helped. I expect author's next work. 

Author Response

Reviewer 1.
In the past, insects appearing in art of Medieval Europe and ancient Egypt have been often studied (for example, Nazari (2014) and Nazari & Evans (2015)). In contrast, analysis subject of this paper is mainly modern art. Generally this paper has high originality.
Thank you for recognizing the highly original nature of our paper. This study is not an overview of insects in art, but, as the title states, about insects in art during a specific period – the environmentally tumultuous one we presently find ourselves dealing with. Because of this, we do not review ancient or medieval examples of insect art. Artistic responses to human-induced environmental disaster is a recent phenomenon, so will be found exclusively in modern and contemporary art. If the reviewer knows of any earlier example, we would be eager to learn about it.
Authors can cite important references and offer appropriate discussion, tables, and figures.
This comment is too vague. We cite what we feel are the most important, relevant references throughout. If the reviewer is referring to historical records of insects in art, please see our comment above.
As authors mentioned in Line 501-504, many arts which are listed are produced in Western Culture. Especially, China is a historic artistic big power, but Asian arts are not discussed much. This bias is a weak point of this paper. It is desirable to do comparative analysis between Western and other regions if possible. However, language barriers cannot be helped. I expect author's next work.
We acknowledge our dearth of representation from several regions of the world (section 4.2), and are fully aware of the wealth of insect art produced in Asia. Aside from the few examples of art by Asian artists we include (e.g., by Asuka Hishiki, Jane Kim, Liao Wenfeng, Vera Ming Wong), we could find no evidence of insect art produced in Asia that is explicitly relevant to our theme. Kazuo Kadonaga, Xu Bing, Yukinori Yanagi, and other important, prolific insect artists have not written, to our knowledge, that their art is motivated by human-induced environmental destruction. If the reviewer knows of any insect art produced in Asia that is relevant to the topic of human-induced environmental destruction, we would be eager to include examples.

Reviewer 2 Report

The manuscript shows the artworks of 73 artists, but it is unclear if a method was applied in the choice of the given examples. Is the criterion a fixed period? or a definite form of art above all of them? Or were the examples randomly picked?

Furthermore, the weight of these works on the large public was not tested at all. Besides, was the aim of the artists to reach a large audience showing the issue of endangered species, or it was not?

Although their paper is not strictly a scientific one, the authors must nevertheless follow the taxonomic rules: often the scientific species names are not in italics (e.g., in the captions), while the common names are often quoted in italics. This is incorrect, and the text must be thorough checked.

The author name(s) must be added to species name the first time that it is quoted.

It is unclear if the artists were aware of the taxonomic position of the subject of their artwork, thus it is uncertain if their choice is meaningful, i.e. it had a real purpose, or it was merely whimsical. It is not wholly clarified.

The iconographic part is more larger than the text part, but the artworks are not wholly described. Perhaps more information in the appendix could be useful.

I surely suggest a major revision, to improve the manuscript. The aim of the paper should surely be made clearer. Also the artworks'presentation should be modified.

Author Response

Reviewer 2.
The manuscript shows the artworks of 73 artists, but it is unclear if a method was applied in the choice of the given examples. Is the criterion a fixed period? or a definite form of art above all of them? Or were the examples randomly picked?
Every work of art that we could find by the means listed in our Materials and Methods section were used to locate examples in which insects or insect products were featured to convey ideas related to human-induced environmental destruction. The period is not fixed, but by the nature of the survey, as discussed above, environmental art is typically modern or contemporary.
To clarify, we now open the Results section with the following: “Environmental art is typically modern or contemporary, so given the nature of our survey, none of our results feature insect art predating the recent environmental movement.”
Lines 514-516: We also add the following to our Qualifiers of survey section: “The environmental art movement is relatively recent, but individual artists’ concerns for biodiversity or concerns about our mistreatment of the planet may predate this movement, and older, relevant insect art may exist.”
We selected all relevant visual art (i.e., not music, etc.). Art was not selected randomly, but examples listed in Table A1 are representative works by the artists that were relevant, as we state in Materials and Methods (excerpt):
“We included artists in our survey if descriptions of at least one of their insect works was explicitly relevant to the theme, as expressed by the creators of the art, or by authors, journalists, or critics writing about the art. If the artist produced multiple relevant works, we selected representative pieces that maximized the number of different categories of human-induced environmental distress or the number of insect orders featured.”
We now add the following, from Materials and Methods, to the caption for Table A1 to clarify this:
“If the artist produced multiple relevant works, we selected representative pieces that maximized the number of different categories of human-induced environmental distress or the number of insect orders featured.”
Furthermore, the weight of these works on the large public was not tested at all.
This was not the objective of our study. This would make for a fascinating follow up study.  
Besides, was the aim of the artists to reach a large audience showing the issue of endangered species, or it was not?
Motivations by artists differ, as we state in our paper. We did not extend our interpretations about artists’ motivations beyond what has been explicitly recorded, or what the artists told us directly. We felt that would be over-reaching and assume too much.

Although their paper is not strictly a scientific one, the authors must nevertheless follow the taxonomic rules: often the scientific species names are not in italics (e.g., in the captions), while the common names are often quoted in italics. This is incorrect, and the text must be thorough checked.
Thank you for pointing this out. We have gone back and italicized species names, as is required by convention. The exceptions are found when the text (e.g., title of an art piece; Line 368) is italicized and includes a species name. When this happens, the convention is to have species’ names be the only part of the text that is not italicized. 

The author name(s) must be added to species name the first time that it is quoted.
We have made this change, also adding year (as well as parentheses, when appropriate) throughout the body of the text (Figure legends 1,3,4,6, and the first mention of Apis mellifera). We did not include authors’ names in Table A1 due to space constraints.
It is unclear if the artists were aware of the taxonomic position of the subject of their artwork, thus it is uncertain if their choice is meaningful, i.e. it had a real purpose, or it was merely whimsical. It is not wholly clarified.
This varied by artist. To conduct a thorough analysis of intent to use one species or one order versus another could be the subject of an additional study. Choices were rarely whimsical when it came to the category of “decline of insect pollinators,” because the artists were clear about their intent to identify actual pollinators and their concern for them. Beyond that, intentionality of taxon selection varied from piece to piece, and was beyond the scope of this study.
The iconographic part is more larger than the text part, but the artworks are not wholly described. Perhaps more information in the appendix could be useful.
We are not sure we understand what additional information is desirable here. If the reviewer is suggesting we include art media used, etc., we like the idea, but there are two reasons why it might not be advisable to do so:
1. Including art medium, etc. would only apply to the representative piece included in the table, which would not necessarily extend to other relevant examples by the same artist. The meaningfulness might be lost.
2. Table A1 is already long.
If the editor concurs with the reviewer in light of the two points above, we would be happy to include additional information (e.g., art medium used in representative pieces).
I surely suggest a major revision, to improve the manuscript. The aim of the paper should surely be made clearer. Also the artworks'presentation should be modified.
We believe we are sufficiently clear throughout our manuscript with respect to our aims, as repeated in the title, abstract, introduction, and every other section of the paper.
Please let us know how the artworks’ presentation should be modified, if not in the manner alluded to above (art medium listed in Table A1).

Additional changes made:
Lines 9-11: Changed “eliciting” to “affecting,” and changed “evoking emotions” to “eliciting responses.”

Line 443: Changed “group” to “order” (because a more inclusive grouping, like Class, would obviously be more speciose than an Order contained within it).

Line 453: Added “on insect ordinal abundances” to specify which data are lacking.

Changed year of publication for Wilson’s Future of Life (2002 vs. 2003).